# Extracellular Vesicles Secreted by *Acanthamoeba culbertsoni* Have COX and Proteolytic Activity and Induce Hemolysis

**DOI:** 10.3390/microorganisms11112762

**Published:** 2023-11-14

**Authors:** Francisco Sierra-López, Ismael Castelan-Ramírez, Dolores Hernández-Martínez, Lizbeth Salazar-Villatoro, David Segura-Cobos, Catalina Flores-Maldonado, Verónica Ivonne Hernández-Ramírez, Tomás Ernesto Villamar-Duque, Adolfo René Méndez-Cruz, Patricia Talamás-Rohana, Maritza Omaña-Molina

**Affiliations:** 1Laboratory of Amphizoic Amoebae, Faculty of Superior Studies Iztacala, Medicine, National Autonomous University of Mexico (UNAM), Tlalnepantla 54090, Mexicoismaelc.40@gmail.com (I.C.-R.); mdhm_65@iztacala.unam.mx (D.H.-M.); seguracd@unam.mx (D.S.-C.); armendez@unam.mx (A.R.M.-C.); 2Department of Infectomics and Molecular Pathogenesis, Center for Research and Advanced Studies, National Polytechnic Institute (IPN), Mexico City 07360, Mexico; lsalazar@cinvestav.mx (L.S.-V.); vhernandezr@cinvestav.mx (V.I.H.-R.); ptr@cinvestav.mx (P.T.-R.); 3Department of Physiology, Biophysics and Neurosciences, Center for Research and Advanced Studies, National Polytechnic Institute (IPN), Mexico City 07360, Mexico; ceflores@fisio.cinvestav.mx; 4General Biotery, Faculty of Superior Studies Iztacala, Biology, National Autonomous University of Mexico (UNAM), Tlalnepantla 54090, Mexico; vidutoer@yahoo.com.mx

**Keywords:** *Acanthamoeba culbertsoni*, extracellular vesicles, pathogenicity mechanisms, COX

## Abstract

Several species of *Acanthamoeba* genus are potential pathogens and etiological agents of several diseases. The pathogenic mechanisms carried out by these amoebae in different target tissues have been documented, evidencing the relevant role of contact-dependent mechanisms. With the purpose of describing the pathogenic processes carried out by these protozoans more precisely, we considered it important to determine the emission of extracellular vesicles (EVs) as part of the contact-independent pathogenicity mechanisms of *A. culbertsoni*, a highly pathogenic strain. Through transmission electronic microscopy (TEM) and nanoparticle tracking analysis (NTA), EVs were characterized. EVs showed lipid membrane and a size between 60 and 855 nm. The secretion of large vesicles was corroborated by confocal and TEM microscopy. The SDS-PAGE of EVs showed proteins of 45 to 200 kDa. Antigenic recognition was determined by Western Blot, and the internalization of EVs by trophozoites was observed through Dil-labeled EVs. In addition, some EVs biological characteristics were determined, such as proteolytic, hemolytic and COX activity. Furthermore, we highlighted the presence of leishmanolysin in trophozites and EVs. These results suggest that EVs are part of a contact-independent mechanism, which, together with contact-dependent ones, allow for a better understanding of the pathogenicity carried out by *Acanthamoeba culbertsoni*.

## 1. Introduction

Free-living amoebae of *Acanthamoeba* genus are cosmopolitan protozoa that are ecologically relevant in maintaining a bacterial population balance worldwide [1]. It is important to highlight that some species of the genus are potential pathogens and etiological agents of different diseases in humans as well as in vertebrates. For this reason, they have been more appropriately called amphizoic amoebae, a term that better defines them, in which the exozoic phase corresponds to the free-living stage and the endozoic phase to the parasitic life, during which they behave as opportunistic pathogens [2]. These amoebae are etiological agents of several pathologies, such as amoebic keratitis (AK), granulomatous amoebic encephalitis (GAE), and cutaneous acanthamebiasis pathologies that are difficult to diagnose and resolve because there is no treatment of choice [3,4,5,6,7].

In vivo, in vitro, and ex vivo studies have been implemented with the purpose of elucidating and describing the pathogenic mechanisms that carry out these protozoans when coming into contact with the target tissue, which has allowed us to show these processes in the cornea, central nervous system (CNS), skin, and other organs, evidencing the relevant role of contact-dependent mechanisms [8,9,10,11], which implies the adhesion of the trophozoites to the target tissue. In this process, several molecules have been described: a 400 kDa mannose-binding protein formed by 130–134 kDa subunits [12,13], a 207 kDa surface membrane glycoprotein [14], and the 27–29 kDa laminin-binding proteins (LBPs) [12] and the 55 kDa laminin–collagen-binding protein described in *Acanthamoeba culbertsoni* [15]. In addition, a putative Leishmanolysin gene of *Acanthamoeba castellanii* str. Neff (XP_004337275.1), a homologous protein to Leishmanolysin, GP63 from *Leishmania*, with adhesin [16], protease, and cyclooxygenase (COX) activities [17], has been described. Once firmly adhered, the amoebae migrate towards the cell junctions, and by mechanical and/or enzymatic effects trophozoites, penetrate gradually, introducing their acanthopods between the intercellular junctions [8,18,19], until invading and altering deeper layers and the target tissue architecture [9,10], without inducing the lysis or cytopathic effect of surrounding cells, or even without destabilizing its cytoskeleton [8]. Subsequently, through the emission of different structures [19], which range from large food cups [20] to fine and small cylindrical structures (sucker-like structures) [21], the amoebae phagocytose the detached cells.

As a part of contact-independent mechanisms, it has been shown that *Acanthamoeba* spp. releases hydrolytic enzymes, such as cysteine proteases, elastases, and phospholipases [22,23,24] as well as serine proteases, that induce the degradation of the tight junction proteins ZO-1 and occludin [25]; collagen type I, III [26] and IV; elastin; and fibronectin [27]. These proteases could facilitate the invasion of amoebae through cell junctions, without causing their destruction [9]. Furthermore, Chávez-Munguía et al., in 2016, described that *A. culbertsoni* trophozoites, a highly virulent strain isolated from a clinical case of AK, produces electron-dense granules, which induced the separation of tight junctions when they interacted in vitro with Madin-Darby Canine Kidney (MDCK) cells [28].

Recently, the role played by *Acanthamoeba* spp. extracellular vesicles (EVs) in the pathogenic process have been studied. EVs are spherical particles delimited by a lipid bilayer with a diameter between 50 and 5000 nm, which are released into the extracellular medium by prokaryotic and eukaryotic cells, that stand out as part of the pathogenic mechanisms [29,30]. Particularly, in *Acanthamoeba* [31,32,33,34], it has been reported that EVs induced adverse effects on host cells, such as enzymatic activity, the induction of cell death by apoptosis and/or necrosis, or the induction of immunomodulatory effects. Until now, the emission of *A. culbertsoni* EVs has not been reported. The purpose of this study was to isolate and describe some EVs characteristics emitted by *A. culbertsoni* (ATCC 30171), a highly virulent strain [35], isolated from clinical cases, thereby providing information concerning the proteolytic, hemolytic, and COX activity of these EVs.

## 2. Materials and Methods

### 2.1. Amoeba Culture

*Acanthamoeba culbertsoni* (ATCC 30171) a highly virulent strain, originally isolated from primary monkey kidney tissue culture, was cultured in axenic PYG medium, highlighting that this culture medium does not require fetal bovine serum. The medium was filtered with a 0.22 µm membrane to ensure that it was free of particles before each assay. Cultures were incubated at the optimal temperature of growth of 30 °C. The strain under study was harvested in the exponential phase of growth, during which cultures were chilled at 4 °C for 10 min. Then, trophozoites were concentrated by centrifugation at 1100× *g* for 10 min and washed 2 times with PBS, until use in subsequent assays. 

### 2.2. Extracellular Vesicle (EV) Isolation

The experimental strategy carried out in this study was performed according to the guidelines of Minimal Information for Studies of Extracellular Vesicles (MISEV2018) [36]. The strain in study was cultured in PYG medium (75 cm^2^ flasks) until reaching 90% confluency cultures, and the medium was replaced with 10 mL of modified PYG medium that was peptone and yeast-extract free, according to Gonçalves et al., 2018 [33] and incubated for 48 h at 37 °C (human body temperature). Then, the culture’s supernatant was recovered to purify the EVs for subsequent assays according to the methodology described below.

The supernatants were collected and centrifuged at 1100× *g* for 10 min and subsequently filtered with 1.2 µm membranes to eliminate trophozoites and debris in suspension. Then, samples were centrifuged at 16,800× *g* for 30 min at 4 °C (rotor FA-45–18-11. Eppendorf^®^, Hamburg, Germany), and the supernatant was decanted. Then, EVs were washed with PBS twice and were finally resuspended in 100 µL of PBS. The entire process was carried out in a cold room.

### 2.3. EVs Characterization

The EVs obtained were characterized by transmission electron microscopy (TEM) using negative staining as well as nanoparticle tracking analysis (NTA).

#### 2.3.1. TEM 

Negative staining was performed according to the standard method. The secreted vesicles (5 μL) were pipetted onto the surface of the formvar-coated copper grids (400 mesh). Samples were blotted off with filter paper and stained with 2.5% uranyl acetate for 20 s. Grids were left to air dry and carbon-coated in a vacuum evaporator (JEE400, JEOL Ltd., Tokyo, Japan). Samples were examined using a JEM-1011 transmission electron microscope with Gatan, Orius SC1000A1 camera, and Gatan Digital Micrograph Version 2.30.542 software.

#### 2.3.2. NTA

To perform nanoparticle tracking analysis, the EVs were diluted at 1:500 in filtered PBS (0.22 µm), and their size distribution was determined using a Nanosight NS300 equipped with a sCMOS camera and NTA 3.2. Dev Build 3.2.16 software. The detection threshold was set to 8, the camera level was set to 13, and the blur and Max Jump distance were set automatically. Measurements were performed in triplicate at a temperature of 25 °C.

### 2.4. EVs Emission

To obtain samples containing both EVs and trophozoites, emphasizing the possible observation of the moment of emission of these EVs, several assays were carried out.

#### 2.4.1. Confocal Microscopy

To observe EVs by confocal microscopy, 2 × 10^6^ trophozoites were placed on specific coverslips, which were placed on 6-well plates with a final medium volume of 2 mL and incubated for 2 h to promote the adhesion and emission of vesicles. 

Subsequently, EVs and trophozoites were fixed with 4% paraformaldehyde in PBS at 30 °C for 1 h and blocked with 10% fetal calf serum in PBS for 1 h. Samples were then incubated at 4 °C overnight with the anti-*A. culbertsoni* serum (1:600 dilution). Subsequently, they were incubated with the goat anti-mouse IgG mAb (H+L) secondary antibody FITC (1:600 dilution) (Invitrogen TM, Thermo Fisher Scientific, Waltham, MA, USA) for 1 h. Finally, a mounting medium with DAPI (ProLongTM Diamond Antifade Mountant, Invitrogen TM, Life Technologies Corporation, Eugene, OR, USA) was used, and the samples were observed using a Carl Zeiss LSM 700 confocal microscope (Carl Zeiss AG, Oberkochen, Germany). Antigen recognition was registered in both trophozoites and EVs. Sera with polyclonal anti-*A. culbertsoni* antibodies (pAb) were obtained in BALB/c mice according to conventional immunization schemes.

#### 2.4.2. TEM

To confirm trophozoites EVs emission and describe their morphological characteristics through TEM, samples were processed as shown in previous sections. The supernatants were collected; subsequently, the samples were centrifuged at 16,800× *g* for 30 min, and fixed with 2.5% (*v*/*v*) glutaraldehyde in 0.1 M sodium cacodylate buffer at pH 7.2, then were post fixed with 1% osmium tetroxide, dehydrated with increasing concentrations of ethanol, transferred to propylene oxide, embedded in Polybed epoxy, and polymerized at 60 °C for 24 h. Ultra-thin sections (60 nm) were obtained which were contrasted with uranyl acetate and lead citrate to be observed using a Jeol JEM-1011 transmission electron microscope (JEOL Ltd., Tokyo, Japan) with a Gatan, Orius SC1000A1 camera and Gatan Digital Micrograph Version 2.30.542 software.

### 2.5. Determination of the Protein Pattern by Electrophoresis and Immunorecognition by Western Blot

Trophozoites and EVs were collected in PBS with a protease-inhibitor cocktail, then lysed by heat shock (5 cycles for trophozoites and 2 for EVs), and β-mercaptoethanol (1:4) was added and boiled for 4 min. Proteins were separated on 12% SDS-PAGE gels, where 20 µg of EV proteins and trophozoite extract were loaded.

To detect EV antigens of *A. culbertsoni* by Western Blot, the proteins from the SDS-PAGE gels were transferred to PVDF transfer membranes for immunodetection. Anti-*Acanthamoeba* polyclonal serum (1:2500 dilution) was used as the primary antibody, and an anti-mouse IgG (H+L), alkaline phosphatase (AP) conjugate (1:5000 dilution) (Invitrogen TM) was used as the secondary antibody. Novex™ AP Chromogenic Substrate (BCIP/NCP) (Life Technologies Corporation, Carlsbad, CA, USA) was used to develop the samples.

### 2.6. Internalization of EVs in A. culbertsoni Trophozoites

To determine if there is intra-species communication by vesicles, the freshly purified EVs were stained with Dil (3 µM), and subsequently, interacted with the strain in study for 30 min at 37 °C. They were then fixed with 4% paraformaldehyde, washed three times with PBS, and mounted. Samples were analyzed using a confocal laser scanning microscope (Leica TCS SP8, Leica Camera, Wetzlar, Germany) and processed with Leica Microsystems CMS GmbH version 3.5.21594.6.

### 2.7. Biological EVs Activity 

#### 2.7.1. Proteolytic Activity

Zymograms were carried out for the analysis of proteolytic activity. EVs were prepared as described previously, and then they were suspended with the standard loading buffer (glycerol, Tris-HCl 0.5 M pH 6.8, SDS). Then, 10 µg of EVs were separated by electrophoresis in 8% standard SDS-polyacrylamide gels (distilled water; acrylamide/bisacrylamide; Tris-HCl, 1.5 M, pH 8.8; SDS; ammonium persulfate; and TEMED) copolymerized with 0.4% porcine skin gelatin (Sigma G2500, Sigma-Aldrich, St. Louis, MO, USA) at 4 °C in no-reductor conditions. SDS was removed by washing the gels twice in 1% Triton X-100 for 30 min for subsequent incubation overnight at 37 °C in a developing buffer (50 mM Tris-HCl, 10 mM CaCl_2_, pH 7.0). Zymograms were stained by Coomassie blue, with non-stained areas indicating proteolytic activity.

A densitometric analysis (semiquantitative) was performed with Multigauge software (2003) comparing the total *A. culbertsoni* trophozoites proteolytic activity versus EVs.

#### 2.7.2. Determination of Hemolysis Activity in EVs

To determine the EVs capability to induce erythrocytes lysis, the experiments were carried out following the instructions according to Pierson et al., 2011 [37]. Briefly, from a 2% (*v*/*v*) blood solution in PBS at pH 7.0, 250 µL were taken, then, 750 µL of PBS with 5 µg of EVs were added and incubated for 4 h at 37 °C. After this time, the samples were centrifuged at 200× *g* for 1 min, and the absorbance of the supernatant was read using a spectrophotometer at a wavelength of 540 nm. Erythrocytes in distilled water were used as the positive control, and erythrocytes in PBS were used as the negative control. With these data, the percentage of hemolysis was calculated using the following formula: $$\left(\frac{Absorbance\;of\;sample-Absorbance\;of\;no\;hemolysis}{Absorbance\;of\;total\;hemolysis-Absorbance\;of\;no\;hemolysis}\right)\times 100\;$$

#### 2.7.3. Determination of Cyclooxygenase Activity in Trophozoites and EVs

With the proposal to determine the presence of trophozoites and EVs’ COX activity, assays were performed using a Chemiluminescent Cyclooxygenase Activity Kit according to the manufacturer’s instructions (Cat. No. 907-003, Enzo Life Sciences, Inc., Farmingdale, NY, USA). The assay took place in a 96-well plate using 50 µg of trophozoite as well as EV extract proteins obtained using the Np-40 detergent. Arachidonic acid (AA) was added to the samples. Records were taken immediately after adding the AA to the Fluoroskan Ascent FL Thermo Lab system (Finland) Specific activity results are expressed as Relative Light Units (RLUs) per microgram of protein [38].

#### 2.7.4. Leishmanolysin-like Protein Detection in *A. culbertsoni* Trophozoites and EVs

To determine whether the amoeba under study produced leishmanolysin-like proteins, that is, proteins with adhesin [16], protease, and cyclooxygenase (COX) activities [17], trophozoites and EVs were processed for confocal microscopy, as previously described, using mouse polyclonal antibody anti-GP63 leishmanolysin-like proteins (1:600) (USBiologid Life Science, Swampscott, MA, USA) as the primary antibody and anti-mouse IgG mAb (H+L) FITC antibody (1:600) (Invitrogen TM) as secondary antibody. Samples were observed using a Carl Zeiss LSM 700 confocal microscope. 

### 2.8. Statistical Analysis

Statistical analyses were performed to compare hemolysis assays and COX activity. GraphPad Prism 5.0 software was used along with the one-way ANOVA test and Tukey’s comparison post-test to compare hemolysis assays, and Student’s *t*-test was used to compare COX activity. A value of *p* < 0.05 and *p* < 0.001 were considered significant.

## 3. Results

### 3.1. Extracellular Vesicles (EVs) Characterization

In this work, *Acanthamoeba culbertsoni* (ATCC 30171) EVs were purified from amoebae cultures following the guidelines of MISEV 2018 [36], which includes differential centrifugations and filtration through a 1.2 µm pore. The EVs obtained were characterized by transmission electronic microscopy (TEM) and nanoparticle tracking analysis (NTA). 

The EVs were obtained from 3 × 10^7^ trophozoites cultures, which were harvested after 48 h at 37 °C incubation in PYG medium free of protease peptone and yeast extract. By using TEM precisely with negative staining, the observation of EVs showed the characteristic morphology: spherical particles delimited by a membrane (Figure 1A). Likewise, through NTA, it was determined that 3 × 10^7^ trophozoites emitted 9.25 × 10^8^ EVs that have a size that ranges between 60 and 855 nm, with a mean size of 173.6 ± 2.1 nm and a mode of 135.2 ± 11.4 nm (Figure 1B,C).

### 3.2. EVs Emission

In order to capture and describe vesicles emission, trophozoites were harvested under the conditions described in the previous paragraph and processed through the confocal microscopy and TEM technique. Through the confocal microscopy images, it was possible to observe spherical particles that were recognized by the polyclonal anti-*A. culbertsoni* antibodies close to the trophozoites (Figure 2A). Likewise, through TEM, it was possible to observe something similar; a spherical particle that presents a lipid bilayer was found near a trophozoite (Figure 2B). These results suggest that it was possible to capture the recent emission of extracellular vesicles.

### 3.3. Protein Pattern by Electrophoresis and Immunorecognition by Western Blot

*A. culbertsoni* EVs were purified and processed to determine the protein patterns through the electrophoresis technique, as well as to determine antigenic recognition with mouse polyclonal anti-*A culbertsoni* antibodies. Once the electrophoresis gel was stained with Coomassie blue, it was observed that the vesicles contained proteins with molecular weights ranging from 45 to 200 kDa. The shift analysis of the trophozoite extracts showed bands with molecular weights between 20 and 260 kDa. Likewise, most protein bands of the vesicles observed by electrophoresis were immunorecognized through Western Blot by the polyclonal antibodies. The nonspecific binding of secondary antibody was not observed (Figure 3).

### 3.4. Internalization of EVs in A. culbertsoni Trophozoites

In order to determine if the EVs were capable of being internalized by the *A. culbertsoni* trophozoites themselves, the recently purified EVs were stained with Dil (3 µM) prior to interaction with the amoebae. It was observed that after 30 min of interaction, the trophozoites internalized the EVs, which were found in amoeba cytoplasm (Figure 4A,B). As a control, trophozoites were processed in the absence of EVs (Figure 4C), which were in optimal conditions. The EV trophozoites internalization may suggest intra-species communication processes.

### 3.5. Biological EVs Activity 

#### 3.5.1. Proteolytic Activity

Proteolytic activity was analyzed by zymograms. EVs and trophozoites exhibited a similar protease profile at pH 7.0 in the region of high molecular weight, specifically in bands of 170 and 260 kDa. However, proteolytic activity was more evident in trophozoite extracts (Figure 5). 

Densitometric analysis of the zymograms demonstrated greater proteolytic activity in trophozoite extracts (36,115 Relative units/µg) than in EVs (26,098 Relative units/µg), which represents 28% less proteolytic activity from the trophozoites.

#### 3.5.2. Hemolysis

To determine if the EVs were able to induce damage to target cells, an interaction between EVs and human erythrocytes was carried out for 4 h at 37 °C. After data processing, it was determined that that EVs induced 23% erythrocyte hemolysis, which is significant compared to the control erythrocytes (Figure 6A,B). Damage to blood cells suggests the participation of EVs in contact-independent pathogenic mechanisms.

#### 3.5.3. Determination of Cyclooxygenase Activity in Trophozoites and EVs

COX activity was quantified using a direct method with an enzymatic reaction that generates chemiluminescence produced by the addition of arachidonic acid, thus obtaining an index of the catalytic activity of COX in real time [38]. In our results, *A. culbertsoni* trophozoites as well as EVs (Figure 7), showed cyclooxygenase activity, however, this activity was significantly greater in extracellular vesicles than in trophozoites.

#### 3.5.4. Leishmanolysin-like Protein Detection in *A. culbertsoni* Trophozoites and EVs

To determine whether amoebae are capable of producing leishmanolysin-like proteins, *A. culbertsoni* trophozoites and EVs were labeled with the anti-leishmanolysin antibody and processed for confocal microscopy. The micrographs highlight the presence of protein throughout the cell body, and the image suggests a greater signal of leishmanolysin in the pseudopodia of the trophozoites. Similarly, it was possible to observe the recognition of particles that suggest the presence of EVs near *A. culbertsoni* trophozoites. The nonspecific binding of secondary antibody was not observed (Figure 8).

## 4. Discussion

The amphizoic amoebae of the *Acanthamoeba* genus, are opportunistic pathogens of medical importance due to the pathologies they cause, for which, up to now, there has not been a treatment of choice. The study of the pathogenic mechanisms of these amoebae can open new perspectives for alternative treatment therapies design.

The study of extracellular vesicles (EVs) emitted by numerous eukaryotic and prokaryotic cells is a research area that has recently generated great interest in the scientific world due to their participation in intercellular communication processes, the modulation of the environment, and, in the case of pathogens, its participation in virulence transfer factors having been described [39,40]. The EVs emission by several extracellular parasitic protozoa has been reported, such as *Trichomonas vaginalis* [41], *Giardia intestinalis* [42], *Entamoeba histolytica* [43], and *Naegleria fowleri* [44,45,46], and although previous studies have carried out assays with EVs emitted by *Acanthamoeba castellanii* [32,33], as well as various genotypes of *Acanthamoeba* genus [31,34], to date, the emission of EVs in *A. culbertsoni* (ATCC 30171), a very virulent protozoan and etiological agent of infections in the central nervous system (CNS), has not been described.

In this study, EVs were obtained from *A. culbertsoni* trophozoite cultures supernatants, which were incubated for 48 h at 37 °C in a modified PYG medium free of peptone and yeast extract, as reported by Goncalves et al., 2018 [33]. The EVs obtained showed the general characteristics that define them: a tendency to be spherical, a membrane that delimits them, and, mostly, a size between 60 and 855 nm [47,48]. It is important to highlight that despite purifying the EVs with a centrifugal force of 16,800× *g*, their average size was recorded within the ranges previously reported in other species of *Acanthamoeba* genus, in which exosome isolation kits were used [32] in addition to ultracentrifugation [31,33,34], which indicates that our purification method was appropriate, obtaining similar results in terms of the size of the EVs reported by other methods, despite being another strain under study.

Under different microscopy strategies, the moment at which the EVs were possibly released by trophozoites was observed. Through confocal microscopy, EVs were located near trophozoites, which suggests its recent release. In previous studies, Nievas et al., 2018 [41], described the EVs secretion from the protozoan parasite *Trichomonas vaginalis* through antibodies and confocal microscopy, which is a useful method for observing EVs larger than 200 nm. In addition, using transmission electronic microscopy (TEM), similar results were observed, where EVs characteristic lipid bilayer were located close to the trophozoites. The emission of the EVs could well have occurred through the evagination of the membrane, as has been documented [41,48].

When analyzing the electrophoretic EVs pattern, bands of protein between 45 and 200 kDa were observed, which differs with respect to the protein pattern reported for *A. castellanii* EVs [32], which may be due to different reasons, including inter-species differences, as well as differences in the culture medium used in each assay. Moreover, Visvesvara and Balamuth (1975) [49] had already reported important differences in the electrophoretic patterns found between soluble and particulate antigens of these species, which suggests that there are species-specific characteristics, including the emitted EVs.

Moreover, the Western Blot analysis of EVs produced by *A. culbertsoni* showed a broad recognition of antigenic molecules, some of which could be important as therapeutic targets or diagnostic molecules. Furthermore, some of these molecules could participate in contact-independent pathogenic mechanisms. EVs obtained from virulent *Acanthamoeba* strains stimulate a lower proinflammatory response in macrophages than non-pathogenic strains characterized by lower levels of nitrite, TNFα, and IL-6 [31]. This correlates with the murine model of GAE in both healthy and diabetic mice, describing the early events of the invasion of *A. castellanii* and *A. culbertsoni* reported by Omaña-Molina et al., 2017 [10,11], in which a weak inflammatory process within the first 96 h post inoculation was observed. This reinforces the idea of an immunomodulatory process of these pathogens, which could be associated with the EVs’ cargo.

EVs carry out an important role in intercellular communication mainly due to the ability to transfer their content [48,50]. Upon release, EVs move through the extracellular medium for variable times and distances [51]. Moreover, it has been observed that EVs also play an important role in intra-species communication. The EVs emitted by drug-resistant *Leishmania* strains contain resistance genes that can be transferred to drug-susceptible strains [52]. Additionally, in *Trichomonas vaginalis* an increase in the adherence capacity was observed when EVs secreted by highly adherent strains interact with low-adherent strains [53]. Moreover, EVs secreted by *Entamoeba* trophozoites during the encystment process induce the same effect on trophozoites that are in optimal conditions [54]. In our work, it was observed that *A. culbertsoni* EVs can be internalized by trophozoites of the same species, which suggests an intra-species communication process, which could favor, among others, the transmission of virulence factors between pathogenic and non-pathogenic strains.

Analysis of the pathogenic processes produced by amoebae of *Acanthamoeba* genus has revealed the importance of contact-dependent mechanisms. In this study, we provide information that allows us to describe those independent contact processes that support the understanding of the mechanisms that are carried out globally during the invasion by these amoebae in the different target tissues. Previously, it had been reported that vesicles from different *Acanthamoeba* genotypes showed proteolytic activity [31,34]. In our results, it was possible to verify that the EVs emitted by *A. culbertsoni* also contain proteolytic activity mainly in high molecular weight proteins, which coincides with the results obtained in electrophoresis, where high-molecular-weight protein bands were present, suggesting that these proteases may play a role in pathogenicity mechanisms. 

The proteolytic activity reported in this study supports the idea that EVs obtained could contribute to contact-independent pathogenic mechanisms, which is reinforced by previous studies where *Acanthamoeba* spp. produce metallo [33], serine, and cysteine proteases [31,34], which could participate during the infection process in the different pathologies they cause, facilitating amoebae penetration between the intercellular spaces so that they migrate towards deeper tissue layers, favoring host tissue colonization. In the strain under study, the activity of phospholipase and serine protease has been significantly correlated with its pathogenicity mechanisms, showing differences in the virulence factors it produces, concerning other non-pathogenic *Acanthamoeba* strains [49,55]. In agreement, it has been reported soluble factors obtained from *Acanthamoeba* in the culture medium supernatant showed enzymatic activity that exerts activity against microglial cells [49,55]. It would be interesting to determine if EVs emitted by *A. culbertsoni* play a role in these processes.

Through in vitro assays, it was previously reported that EVs from *Acanthamoeba* strains induce a cytopathic effect in mammalian cells [32,33]. Moreover, in this work, it has been shown for the first time that EVs induce human erythrocyte lysis. The EVs proteolytic activity suggests that enzymatic activity facilitates destruction and amoebic penetration into deeper tissue layers as a part of contact-independent processes.

Cyclooxygenase (COX), an enzyme that acts on arachidonic acid (AA), is another protein that has been associated with pathogenic mechanisms in parasites to synthesize prostaglandins (PGs), which are important molecules in the inflammatory process in mammals. The presence of COX activity and PGs in the host microenvironment during interaction with different parasites has been reported. PGs can be released by host cells, and it has been suggested to enable pathogen invasion [56,57]. Furthermore, in recent years, COX activity has been determined in different parasites, proposing that part of the PGs in the infection microenvironment could be synthesized by the parasites. Hernández-Ramírez et al. (2023), demonstrated COX activity in *Naegleria fowleri*, *Entamoeba histolytica*, *Giardia duodenalis*, *Acanthamoeba castellanii*, and *Trypanosoma cruzi* [17]. The presence of prostaglandins in *Acanthamoeba* spp. had previously been reported [58]. As well, in *A. castellanii*, it was reported that the use in vitro of non-steroidal anti-inflammatory drugs, whose therapeutic target is COX, inhibits its proliferation and encystment [59]. In this work, we reported, for the first time, COX activity in trophozoites and EVs of *A. culbertsoni*. Finding considerably greater activity in EVs suggests that these particles transport COX-like proteins to the extracellular environment, perhaps for communication purposes or as a virulence factor for these amoebae. It is important to determine the role COX as part of the pathogenic processes of *A. culbertsoni*. 

Furthermore, Hernández-Ramírez et al. (2023) reported a D12 mAb against the Gp63 protein of *Leishmania mexicana* monoclonal antibody, immunodetecting a COX-like protein in *A. castellanii* trophozoites [17]. In *E. histolytica*, the leishmanolysin homologue gene (*E. histolytica* MSP-1) has been reported, which encodes a surface metalloprotease involved in the regulation of amoebic adhesion, with additional effects on cell motility, the destruction of cell monolayers, and phagocytosis [60]. In agreement, in our work, it was possible to observe leishmanolysin-like proteins throughout the cell body. It is important to highlight that the areas of greatest intensity were observed in the pseudopodia of the trophozoites, areas of adhesion of these amoebae. The recognition of leishmanolysin-like proteins evidenced by surface labeling indicates similarities with other parasites [17]. It remains to be determined whether COX activity is also part of the functions of the same molecule, as well as enzymatic activity, specifically metalloprotease, and adhesin.

The existence of isoforms of leishmanolysin or the gp63 protein would explain the different functions attributed to this molecule, which would imply an adhesin function, as well as that of metalloprotease and cyclooxygenase-like (COX-like) proteins. Particularly, adhesin and metalloprotease activities have been localized in the plasma membrane [61], and the COX-like activity has been localized in the nuclear membrane [62]. According to our results, we suggest that a large part of the leishmanolysin-like with COX-like activity is secreted as cargo in the EV of the trophozoites, which could explain the decrease in this activity in trophozoites. The high fluorescent reaction observed in trophozoites with the anti-leishmanolysin antibody by confocal microscopy could be due to the adhesin and metalloprotease isoforms of leishmanolysin (gp63) present in the plasma membrane [63].

In the last decade, the study of extracellular vesicles of bacteria, fungi, protozoa, and helminths of medical importance has boomed due to their implications in contact-independent pathogenicity mechanisms. To date, it has been shown that EVs derived from pathogens can increase the adherence and invasion of these organisms to host cells, and being able to damage and induce cell death, favoring the invasion and establishment of infectious diseases. Furthermore, in parallel, the immunomodulatory effect of their EVs has been demonstrated; some have the capacity to suppress the host’s immune response and favor the survival of the pathogen, while others induce an exacerbated inflammatory response and cause damage to host tissues. In addition to the above, it is recognized that EVs participate in intra- and inter-species communication, through which they can transfer virulence factors that allow the pathogen to survive adverse conditions [40,64,65,66] Through this study, we have begun to show the emission of EVs by *A. culbertsoni*, a highly virulent strain involved in cases of CNS infection that shows hemolytic activity, which indicates that these blood cells are a target for both trophozoites and these vesicles. On the other hand, there is evidence that links leishmanolysin-like proteins with those present in other parasites such as *Entamoeba* and *Leishmania* itself, which has been related to adhesion processes, COX activity, and enzymatic activity, specifically metalloprotease, which opens a path for new research.

These results suggest that EVs are a part of contact-independent mechanisms, which are associated with the contact-dependent pathogenicity mechanisms, and provide a better understanding of the pathogenicity mechanisms carried out by *Acanthamoeba* spp.

## Figures and Tables

**Figure 1 microorganisms-11-02762-f001:**
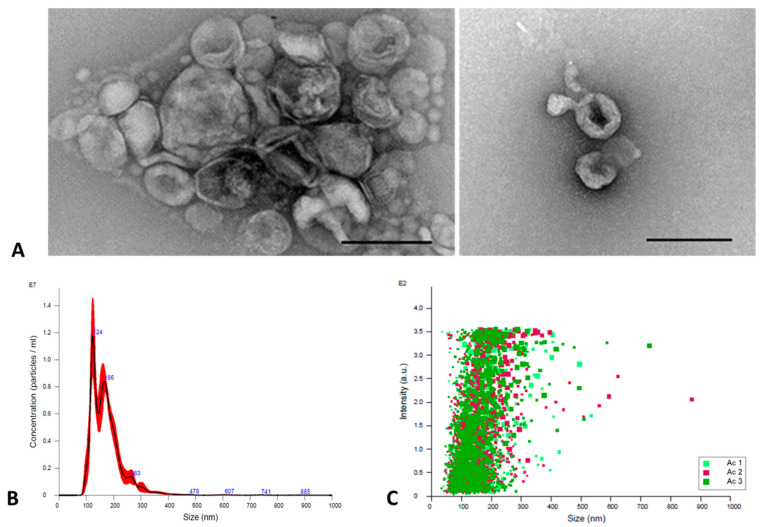
EVs of *A. culbertsoni*. (**A**) TEM. Representative electron micrographs of the spherical EVs delimited by a membrane. Bars = 250 nm. (**B**,**C**) NTA through the Nanosight NS300 equipment. (**B**) Graphic representation of the size and concentration of extracellular vesicles. The average is represented by the black line and the standard deviation by the red line. (**C**) Dot plot of particle size/relative intensity. Assays were performed in triplicate.

**Figure 2 microorganisms-11-02762-f002:**
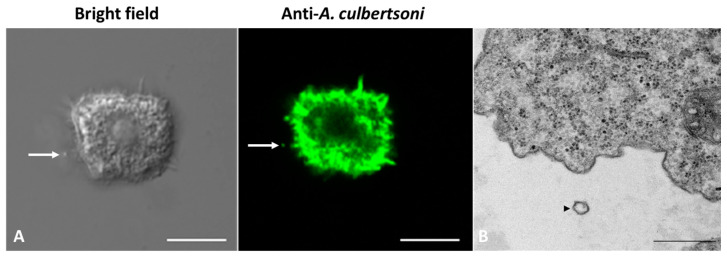
Representative images of the probable secretion of extracellular vesicles by *A. culbertsoni*. (**A**) Confocal microscopy. Spherical particles (arrows) were observed close to a trophozoite, which were recognized by polyclonal anti-*A culbertsoni* antibodies (Bar = 10 µm). (**B**) TEM. It was possible to observe a spherical particle with a lipid bilayer (arrowhead) close to a trophozoite (Bar = 500 nm).

**Figure 3 microorganisms-11-02762-f003:**
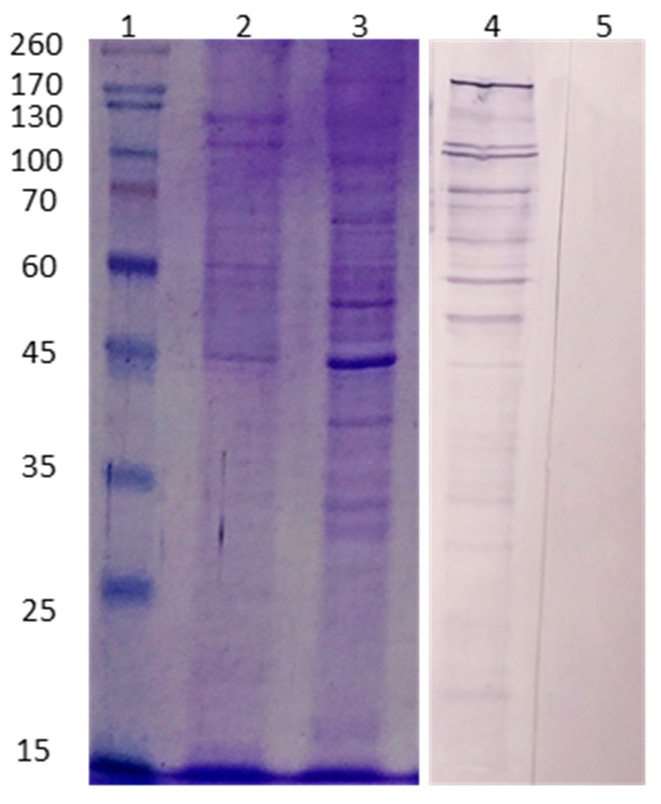
Protein pattern and immunorecognition of *A. culbertsoni* extracellular vesicles. Lane 1–3: electrophoresis. Lane 1: molecular weight marker. Lane 2: extracellular vesicles. Lane 3: trophozoites. Enriched protein bands of between 45 and 200 kDa were observed in the vesicles. Lane 4: Western Blot. Antigenic recognition is observed mainly in protein bands greater than 45 kDa. Lane 5: Nonspecific binding of secondary antibody was not observed.

**Figure 4 microorganisms-11-02762-f004:**
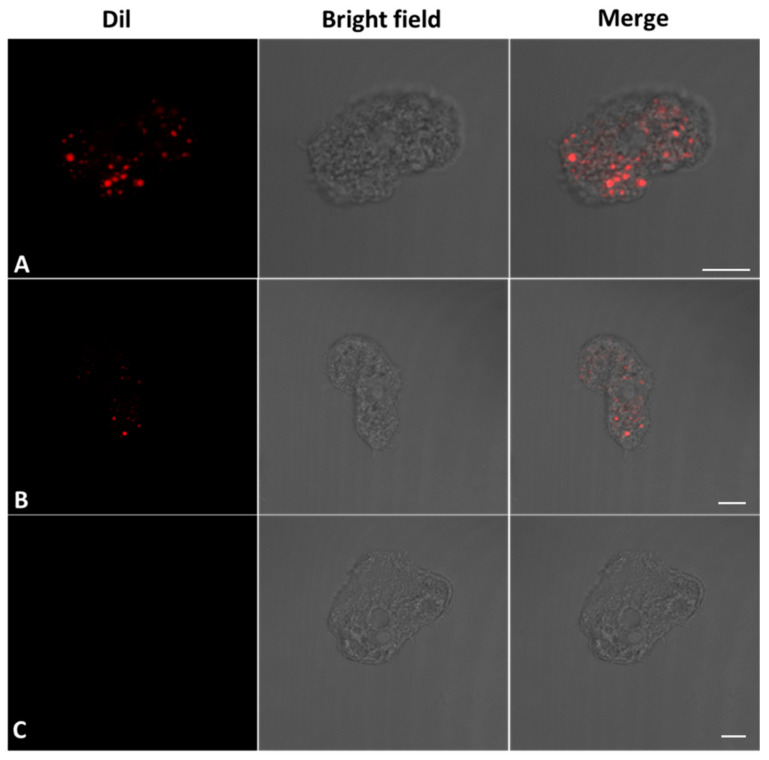
Representative images of the EVs internalization by *A. culbertsoni* trophozoites. EVs were isolated from a trophozoite culture, subsequently stained with DiI, and finally interacted with *A. culbertsoni* trophozoites for 30 min. (**A**,**B**) EVs were observed in the cytoplasm of trophozoites, suggesting intra-species communication. (**C**) Trophozoites in the absence of EVs were used as a control. (Bars = 10 µm).

**Figure 5 microorganisms-11-02762-f005:**
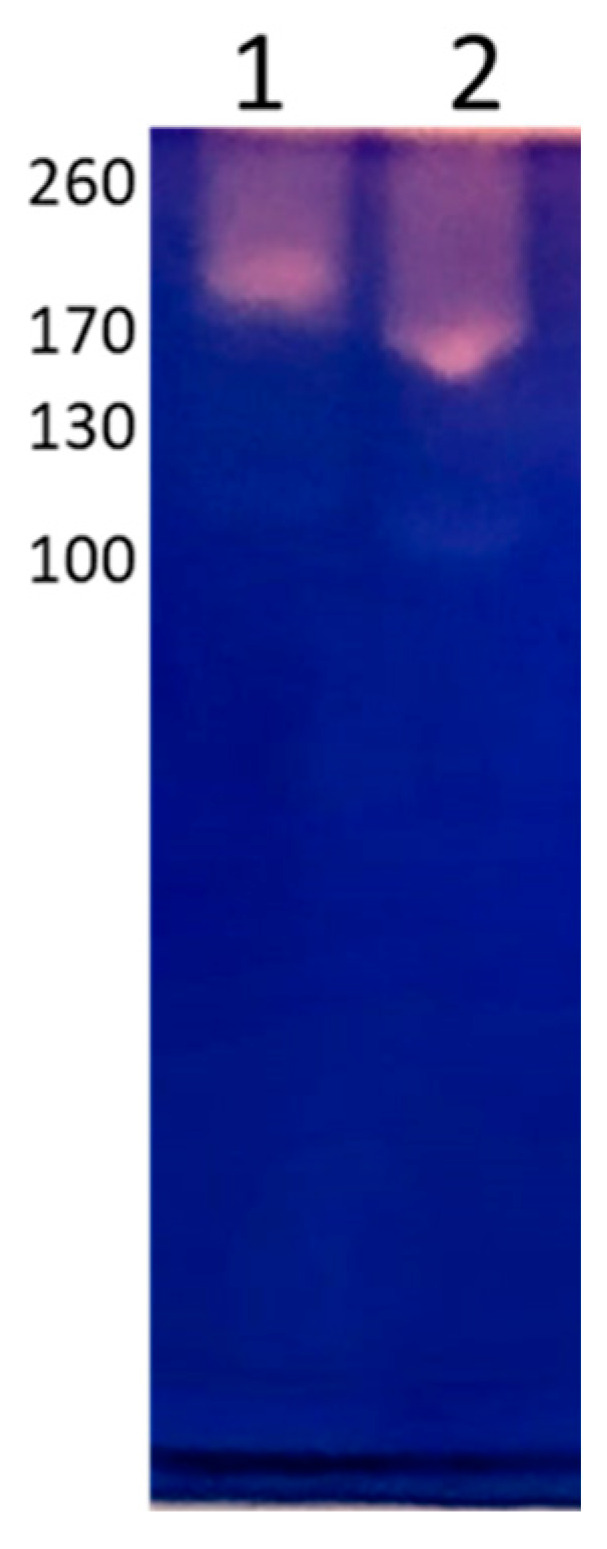
Zymograms. Proteolytic activity was observed in high molecular weight areas, which is more evident in trophozoites. Lane 1: EVs. Lane 2: trophozoites.

**Figure 6 microorganisms-11-02762-f006:**
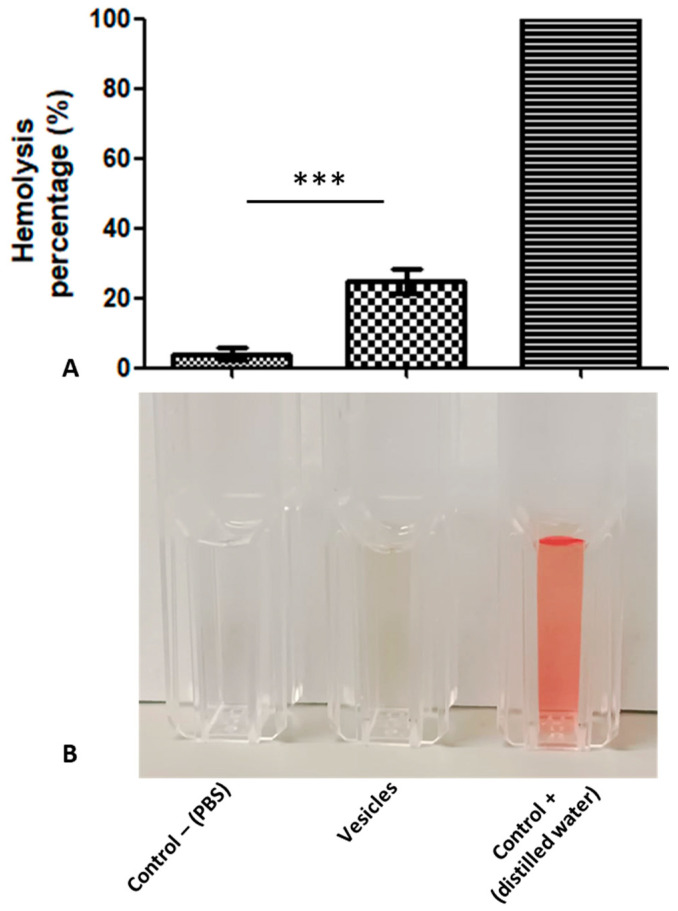
Erythrocytes incubated with *A. culbertsoni* EVs lysis. (**A**) After data analysis, it was found that there was significant erythrocyte lysis in blood cells that interacted with EVs. The average ± standard deviation of the mean is presented (*n* = 3, *** *p* < 0.001). (**B**) Representative image of hemolysis observed.

**Figure 7 microorganisms-11-02762-f007:**
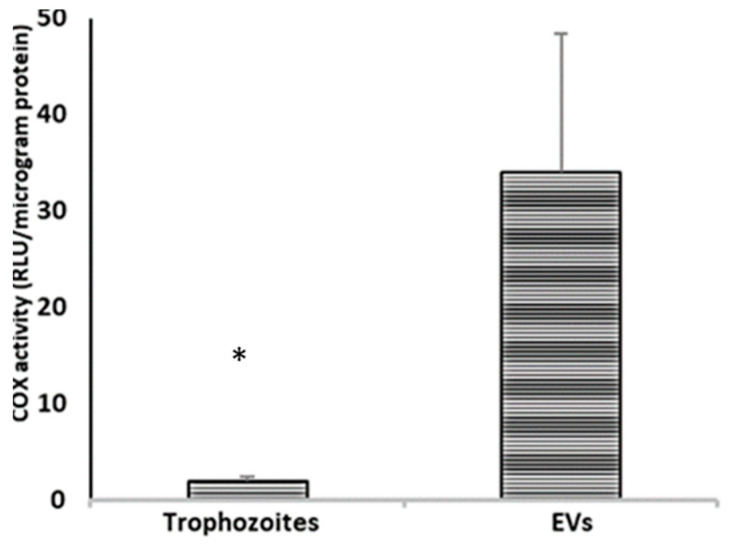
Determination of COX activity in trophozoites and EVs of *A. culbertsoni.* Higher COX activity was observed in vesicles than in trophozoites. The average ± standard deviation of the mean of COX-like activity is presented (*n* = 3, * *p* < 0.05).

**Figure 8 microorganisms-11-02762-f008:**
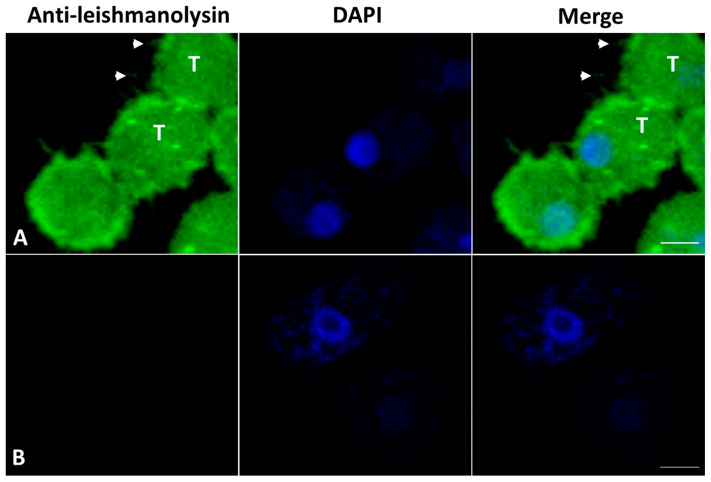
(**A**) Representative image detection of a homologous protein to leishmanolysin (leishmanolysin-like). Protein was detected in trophozoite (T). Similarly, particles that suggests EVs (arrowheads) are shown to be recognized by anti-leishmanolysin antibody, which were near the trophozoites. (**B**) Nonspecific binding of secondary antibody was not observed. Bars = 5 µm.

## Data Availability

Data supporting this study are described in the text.

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
