# Peer review of "Extracellular Vesicles Secreted by Acanthamoeba culbertsoni Have COX and Proteolytic Activity and Induce Hemolysis"

_microorganisms, 2023, doi:10.3390/microorganisms11112762_

Round 1

Reviewer 1 Report

Comments and Suggestions for Authors

In the manuscript “Extracellular vesicles secreted by Acanthamoeba culbertsoni have COX and proteolytic activity and induce hemolysis” Authors investigated some functionalities of A. culbertsoni EVs. This is an interesting and valuable approach, especially with regard to host-pathogen interactions.

However, I have some comments on the manuscript that I would like to ask the authors to consider.

First, the grammar in the manuscript needs to be corrected.

Materials and methods:

The camera and software used for TEM should be provided.

NTA - software and measurement parameters should be included (camera level, threshold etc.).

Results:

Figure 1 - a TEM image showing more structures (wider field of view) is necessary to include to this figure.

What is the efficiency of EVs isolation? How many particles did the initial number of 3 x 10^7 trophozoites produce?

Figure 3 - there is no control for nonspecific binding of secondary antibodies.

In addition to this general characterization of protein content, protein identification is necessary - do the authors have proteomics and mass spectrometry data? This is the major comment to the manuscript - there is definitely a lack of proteomic characterization that could also confirm the presence of the GP63 protein in EVs.

Figure 4 - For each cell, was the number of visible internalized EVs so low? This result is rather inconspicuous, and the merge image suggests a higher signal observed from Dil than shown in the middle image.

Proteolytic activity should be confirmed also with protein/peptide substrate, fluorogenic or colorimetric.

How can the authors explain such a significant difference in COX activity between trophozoites and EVs? Especially if the staining in the next figure shows protein abundance in both locations.

Figure 8 - the photographs are inconclusive, the intensity is similar, and there is no certainty that the structures marked with arrows are EVs without corresponding TEM; there is no control for nonspecific binding of secondary antibodies,

The results are promising, but they require refinement and additional controls.

Comments on the Quality of English Language

The manuscript requires language improvement.

Author Response

We deeply appreciate all suggestions made by you; the time spent for the thorough review.

We made the requested modifications, which will surely enrich the manuscript. according to the suggestions.

We are ready to receive your comments.

Reviewer 2 Report

Comments and Suggestions for Authors

The author in manuscript "Extracellular vesicles secreted by Acanthamoeba culbertsoni. The manuscript in present form needs minor revision for following points:

1. What is the control - in the Figure 6A? What is the control + in the Figure 6A?

2. The authors should point out which is A. culbertsoni trophozoites or EVs in Figure 8?

Comments on the Quality of English Language

Minor editing of English language required

Author Response

We deeply appreciate all suggestions made by you; the time spent for the thorough review.

We made the requested modifications, which will surely enrich the manuscript. according to your suggestions.

We are ready to receive your comments.

Round 2

Reviewer 1 Report

Comments and Suggestions for Authors

I would like to thank the authors for making changes in response to comments on the manuscript and improving its quality.

line 31 – it should be “Dil labeled EVs” instead of “marked”

lines 296, 366, 375 – it should be “was not observed” instead of “was not recorded”

The previous comment on the zymogram and proteolytic activity assay referred to quantitation using a fluorogenic or synthetic substrate in solution. Can the authors perform one with EVs? Additionally in 2.7.1 the detailed composition of the loading buffer and gels should be provided.

Comments on the Quality of English Language

The authors have introduced numerous changes and language corrections, and their verification is necessary in the final version of the manuscript.

Author Response

I would like to thank the authors for making changes in response to comments on the manuscript and improving its quality.

line 31 – it should be “Dil labeled EVs” instead of “marked”

lines 296, 366, 375 – it should be “was not observed” instead of “was not recorded”

We appreciate the suggestions for changes, which have been made in the manuscript.

The previous comment on the zymogram and proteolytic activity assay referred to quantitation using a fluorogenic or synthetic substrate in solution. Can the authors perform one with EVs? Additionally in 2.7.1 the detailed composition of the loading buffer and gels should be provided.

We provided the composition of the loading buffer and gels, which has been added in the methodology section. In addition, we determined the total proteolytic activity of trophozoites and EVs through a semiquantitative densitometric analysis (Multigauge software, 2003 version). The data was aggregated in methodology, results, and discussion. We plan in the future to carry out the proteolytic characterization of the EVs.